# Coral mortality induced by the 2015-2016 El-Niño in Indonesia: the effect of rapid sea level fall

Eghbert Elvan Ampou[1,2,3], Ofri Johan[4], Christophe E. Menkes[5], Fernando Niño[3], Florence Birol[3], Sylvain Ouillon[3], Serge Andréfouët[1,2]

[1]UMR9220 ENTROPIE, Institut de Recherche pour le Développement, Université de la Réunion, CNRS, B.P.A5, 98848, Noumea ,New Caledonia

[2]Institute for Marine Research and Observation. SEACORM / INDESO center. Jl. Baru Perancak, Negara-Jembrana, Bali 82251, Indonesia

[3]Laboratoire d'Etudes en Géophysique et Océanographie Spatiales, Université de Toulouse, CNRS, IRD, CNES, UPS, 14 avenue Edouard-Belin, 31400 Toulouse, France

[4]Research and Development Institute for Ornamental Fish Culture, Jl. Perikanan No. 13, Pancoran Mas, Kota Depok, Jawa Barat 16436, Indonesia

[5]Laboratoire d'Océanographie et du Climat: Expérimentations et Approches Numériques, Sorbonne Universités, UPMC Université Paris 06, IPSL, UMR CNRS/IRD/MNHN, B.P.A5- 98848, Noumea, New Caledonia

*Correspondence to*: Serge Andréfouët,. E-mail: serge.andrefouet@ird.fr

**Abstract:** The 2015-2016 El-Niño and related ocean warming has generated significant coral bleaching and mortality worldwide. In Indonesia, first signs of bleaching were reported in April 2016. However, this El Niño has impacted Indonesian coral reefs since 2015 through a different process than temperature-induced bleaching. In September 2015, altimetry data shows that sea level was at its lowest in the past 12 years, affecting corals living in the bathymetric range exposed to unusual emersion. In March 2016, Bunaken Island (North Sulawesi) displayed up to 85% mortality on reef flats dominated by *Porites, Heliopora* and *Goniastrea* corals with differential mortality rates by coral genus. Almost all reef flats showed evidence of mortality, representing 30% of Bunaken reefs. For reef flat communities which were living at a depth close to the pre-El Niño mean low sea level, the fall induced substantial mortality likely by higher daily aerial exposure a least during low tide periods. Altimetry data was used to map sea level fall throughout Indonesia, suggesting that similar mortality could be widespread for shallow reef flat communities, which accounts for a vast percent of the total extent of coral reefs in Indonesia. The altimetry historical records also suggest that such event was not unique in the past two decades, therefore rapid sea level fall could be more important in the dynamics and resilience of Indonesian reef flat communities than previously thought. The clear link between mortality and sea level fall also calls for a refinement of the hierarchy of El Niño impacts and their consequences on coral reefs.

**Key-words:** ENSO; Absolute Dynamic Topography; Sea Level Anomaly, Coral reef; Indonesia; Coral Triangle

## 1. Introduction

El Niño-Southern Oscillation (ENSO) is the most important coupled ocean-atmosphere phenomenon impacting climate variability at global and inter annual time scales (McPhaden, 2007). The consequences on coral reefs have been well documented, especially since the 1997-1998 massive coral bleaching event, which reached planetary dimension (Hoegh-Guldberg, 1999). In short, El Niño increases temperature in several coral reef regions and induces zooxanthellae expulsion from the coral polyp, resulting in coral colony looking white, hence "bleaching". If the situation persists the coral colony eventually dies. Coral bleaching intensity has been related to different temperature thresholds, other environmental factors and stressors, and type of zooxanthellae and corals (Baker et al., 2008). Bleaching episodes due to ocean warming were recorded during the strong 1982-83 El Niño in Australia (Glynn, 2000) and have since been reported worldwide in several instances (Guest et al., 2012; Wouthuyzen et al., 2015). The last bleaching episode has occurred in 2015-2016 during what occurs to be the strongest El Niño event on record (Schiermeier, 2015). Bleaching events were often global in the past, including Indonesia (Suharsono, 1990; Guest et al., 2012; Wouthuyzen et al., 2015). Last reports for Indonesia in 2016 are still under analysis, and Reef Check survey locations are presented at http://reefcheck.or.id/bleaching-indonesia-peringatan/. It is thus assumed that coral bleaching induced by ocean warming will be the main culprit if post-El Niño surveys report coral mortalities.

While in Bunaken National Park in February 23rd – March 5th 2016 for a biodiversity survey, we noticed recent mortalities on the upper part of many massive colonies on several reef flats. This prompted a systematic investigation of the phenomenon's spatial distribution. We report here observations on what appears to be the first significant impact of the 2015-2016 El Niño on Indonesia reefs. Unlike what is expected during such a strong event, the mortality was not

related to warm water induced-bleaching, but could be tracked to rapid sea level variations. Coral mortality data around
Bunaken Island are provided, and we investigate various altimetry and sea level anomaly data sets to explain mortality.
The clear link between mortality and sea level fall calls for a refinement of the hierarchy of El Niño impacts and their
sequences on coral reefs.

## 2. Material and Methods


Bunaken National Park (BNP) is located at the northwest tip of Sulawesi, Indonesia. The location is at the core of the
epicenter of marine biodiversity, the so-called Coral Triangle, a vast area spanning Malaysia to Solomon Island, where
the number of marine species is maximum (Hoeksema, 2007). BNP includes several islands with Bunaken Island
(1.62379°N, 124.76114°E) one of the most studied Indonesian reef site. Bunaken Island is surrounded by a simple
fringing reef system, comprising reef flats, several small enclosed lagoons and forereefs. The tide regime is semi-diurnal,
but with marked diurnal inequalities (Ray et al. 2005), with a maximum spring tidal range that can reach 2.52 m.
Bunaken Island is generally exposed to southwest wind from May to October, resulting in calm seas due to the short
fetch between mainland and the island, and to northwest wind from November to February, which can be strong at time
and generate large waves breaking on the west and north shores.

Two previous BNP surveys for habitat mapping, in May-June 2014 and May-June 2015, did not show any significant
signs of widespread mortalities on reef flats. Different species of corals were frequently exposed above water level at
low spring tide, yet they were entirely alive (Fig. 1). Microatolls were present. They have not been studied in Bunaken
NP, but by similarity with other sites, their growth is likely constrained by a Mean Low Water (MLW), between Mean
Low Water Neaps (MLWN) and Mean Low Water Springs (MLWS) (Smithers and Woodroffre, 2000; Goodwin and
Harvey, 2008). Several reef flats were characterized by compact communities of massive and semi-massive colonies that
could be described as keep-up communities limited in their vertical growth by the MLW (by analogy with the
terminology of Holocene reefs provided by Neumann and Macintyre, 1985).

In contrast with the 2015 observations, in late February 2016, during a coral biodiversity census survey, we noticed the
widespread occurrences of dead massive corals and we performed a systematic investigation on the spatial distribution of
the phenomenon. Using the habitat map created by Ampou (2016), all coral habitat polygons present on reef flats around
Bunaken Island were visually surveyed and recent mortality was recorded (presence/absence). Geographic coordinates of
the presence of mortality were compiled to map its extent. In practice, when mortality was observed on a habitat
polygon, the entire polygon was flagged as positive. Then, in different locations around the island, mortality was
measured on six reef flat locations characterized by high coral cover and different dominant massive coral species,
principally *Porites lutea* and the octocoral *Heliopora coerulea*, using six 10-meter long Line Intercept Transect (LIT)
(English et al., 1997). We recorded the percent cover of live and dead tissue for each coral and summed the total. We
also recorded the species/genus for each coral, and substrate categories between colonies. We did not keep track onf the
number of colonies present on each transect.

A clear sharp horizontal limit of tissue mortality was present in impacted colonies. The distribution of dead tissue
between colonies and among colonies (Fig. 1) suggested that mortality was related to sea level variations, with increased

aerial exposure time during the last few months. In order to test this hypothesis, we needed to identify sea level variation data. For this, long term data from a tide gauge or a pressure sensor are ideal but these were not available for Bunaken. Tide-gauge data are scarce in Indonesia but fortunately there are two tide-gauges in the north of Sulawesi in the city of Bitung, east of Bunaken, by latitude 1.430N and longitude 125.200E on the other side of Sulawesi compared to Bunaken. Thus, while tide-gauge data are available in the region, they are not exactly on Bunaken, but can help visualize the range of conditions found in Bunaken. Bitung data was retrieved from the Sea Level Center in Hawaii (SLCH), specifically at http://uhslc.soest.hawaii.edu/thredds/uhslc_quality_daily.html?dataset=RQD033A. The Sea Surface Height (SSH) provided is referenced, for Bitung, against a GPS station located at Bako (http://www.igs.org/igsnetwork/network_by_site.php?site=bako) which is itself referenced against the WGS84 ellipsoid. Hence, raw Bitung SSH do not represent absolute depth above the Bitung seafloor. SLCH provides high quality data (available till early 2015) that have been controlled for most outliers and errors, and lower quality data that includes the most recent coverage, included our period of interest (2015-2016). The Bitung tide gauge stopped recording in many instances for raisons unknown to us, hence the records present many, irregularly-spaced, gaps.

In addition to the Bitung tide gauge data, different sea level anomaly products were investigated, based on their temporal coverage and spatial resolution. First, we used gridded altimetry data in terms of Absolute Dynamic Topography (ADT), from the Archiving, Validation and Interpretation of Satellite Oceanographic Data center (AVISO) at the spatial resolution of ¼°. ADT provides the sea level with respect to the geoid. Data is available from 1993 to 2016, allowing a long-term comparison of the sea level trends. The mean ADT over the period were extracted for a small box next to Bunaken Island (1.5-1.7° N; 124.5-124.8° E), a larger box (3 by 3 degrees around the smaller box) centered on Bunaken Island and including the north of Sulawesi and Tomini Bay in the south, and for the entire Indonesia (-14.9-10.0°S, 94.9-140.0°E). The difference between the minimum value (observed in September 2015) and the 2005-2014 mean or the 1993-2016 mean periods were also computed. In addition, we also retrieved ADT data corresponding to the Bitung tide gauge location to compare altimetry sea level anomalies with *in situ* data. The selected retrieved location is the closest available from Bitung (1.375N and longitude 125.125E). To compute sea level anomalies, we considered only the periods of time covered by both data sets in order to use a common baseline.

Second, to extract geophysical information from higher spatial resolution altimeter data, we used the along-track measurements from SARAL/AltiKa Geophysical Data Records (GDRs) distributed by the AVISO service (http://www.aviso.altimetry.fr/fr/). This data set was chosen because the new Ka-band instrument from SARAL has a finer spatial resolution and enables a better observation of coastal zones (Verron et al., 2015). Data extends from March 2013 (cycle 1 of the satellite mission) to May 2016 (cycle 33), with a repeat period of 35 days. Over this period, we use all altimeter observations located between 10°S-10°N and 105°E-140°E. Two tracks (#535 and #578) intersect the north of Sulawesi Island and contain sampling points just off Bunaken Island. The data analysis is done in terms of sea level anomalies (SLA) computed from the 1-Hz altimeter measurements and geophysical corrections provided in GDRs products. The SLA data processing and editing are described in details in Birol and Niño (2015). The 1-Hz SLA data have a spatial resolution of ~7 km along the satellite tracks. In order to quantify the spatial variations of the regional sea level change in March 2013-May 2016, a linear trend model is applied (using a simple linear regression) to the individual SLA time series observed at the different points along the altimeter tracks crossing the area of interest. The trend is the slope of the regression (in $cm.y^{-1}$). The resulting 3-year sea level trend values can be represented on a map.

136

## 3. Results

### 3. 1 Mortality rates per dominant coral genus

For all colonies found on the six stations, dead tissues were found on the top and upper-flank of the colonies, with the lower part of the colonies remaining healthy (Fig. 1). Mortality was not limited to microatoll-shaped colonies only. Round massive colonies were also impacted. On microatolls and other colonies that may have lived close to MLW, the width of dead tissue appeared to be around a maximum of 15 cm. Dead tissues were systematically covered by turf algae, with cyanobacteria in some cases, suggesting that the stressor responsible for the mortality occurred few months earlier. There were no obvious preferential directions in tissue damage at colony surface as it has been previously reported for intertidal reef flat corals in Thailand (Brown et al. 1994).

146

The six surveyed reef flat locations were dominated by *H. coerulea* and *P. lutea*, while other genus and species occurred less frequently (Table 1). When taking into account all genus, up to 85% of the colonies were dead (Site 5). The average mortality was around 58% all sites included (Fig. 2). When it was present *Goniastrea minuta* colonies were the most impacted, with a 82% mortality on average (Fig. 2). Highest mortalities were found on keep-up communities relative to sea level (Fig. 1).

152

### 3.2 Map of occurrences of mortality

The survey around the island revealed presence of mortality all around the island except the north reef flats where corals were scarce and encrusting (Fig. 2). The same coral genus as listed in Table 1 were impacted, but mortality levels differed depending on colony heights. When colonies were clearly below the present minimum sea level, they remained healthy (Fig. 1). Locations of positive observations show that mortality has occurred mostly along the crest, which is expected to be the most vulnerable during sea level fall (Fig. 2). The spatial envelop of mortality occurrences is shown on Figure 2's Bunaken map. The survey and generalization through the habitat map suggests that nearly 163 hectares, or 30% of the entire reef system, has been impacted by some mortality. However, this does not mean that 30% of the reef has died.

162

### 3.3 Comparison between tide gauge and altimetry data

We found a good correlation (Pearson r=0.83) between sea level anomalies from altimetry (ADT) and from tide-gauge. The two time series are compared Figure 3 to confirm the agreement. The Bitung sea-level data reveal the type of sea-level variations that likely occurred around Bunaken, although patterns may not be exactly the same considering the distances between sites. Figure 3 shows from the available Bitung data the daily mean sea level (that can be compared to sea level as provided by altimetry), and the daily lowest level (which can not be directly measured by altimetry). This graph suggests what was likely the range of sea level variations happening in Bunaken before El Niño, due to normal tide fluctuations. The daily lowest value (blue curve in middle and lower panels in Fig. 3) exhibited a ~40-cm variation

from neap tide to spring tide. In 2014, and 2015, we witnessed during spring tide conditions *Porites* corals that had the
upper part of the colonies well above the sea level, and without signs of mortality (Fig. 1). Hence, the upper limit of coral
survival is somewhere around 20 cm above the spring tide lowest level for the end of the period shown on Figure 3. In
other words, the limit of coral survival is close to the mean of the daily lowest level curve. If this mean value is changing
through time, the limit of mortality also changes dynamically. The ~15cm fall that we observed on altimetry data around
Bunaken and on most of east Indonesia changed for a short time (of several weeks) the lowest levels, and these changes
lasted long enough so that coral tissues were damaged by excessive UV and air exposure. During few weeks in August-
September 2015, this fall resulted in a shift of the mean low level towards the pre-El Niño lowest levels shown Figure 3
(lower panel).

**3.4 Absolute Dynamic Topography time series**
The ADT time-series (Fig. 4) shows a significant sea level fall congruent with El Niño periods, at all spatial scales,
although the pattern is not as pronounced at Indonesia-scale (Fig. 4). The 1997-1998 and the 2015-2016 years display the
highest falls. The September 2015 value is the local minima, considering the last ten years (Fig. 4). The 8 cm fall in
September 2015 compared to the previous 4 years, and the 15 cm fall compared to the 1993-2016 mean (Fig. 4) is
consistent with the pattern of mortality following a maximum of ~15 cm width on the top of the impacted micro-atolls
and colonies that were living close to the mean low sea-level before the event (Fig. 1).

**3.5 Sea Level Anomaly trends**
SARAL/AltiKa data in March 2013-May 2016 are shown in Figure 5 for a small area that includes Bunaken Island (top)
and a larger box (bottom) covering part of the western equatorial Pacific Ocean and Coral Triangle. A substantial sea
level fall is observed around Bunaken Island, with values ranging from 4 to 8 cm/year (12 to 24 cm accumulated over 3
years, Fig. 5). Further analysis of the individual sea level time series indicates that the overall trend is explained, and
accelerated, by the fall due to El Niño (not shown). This result agrees with findings from *Luu et al.* (2015) around
Malaysia and can be extended to much of the Coral Triangle. The Figure 5 shows that this phenomenon is consistent
over a large part of Indonesia and the warm water pool, with strong differences in sea level variations (up to -15 cm/year
are observed between Asia and Micronesia, north of 5°N and east of 130°E).

**4. Discussion**
A common ground exists between this study and the use of massive corals to reconstruct sea level. Reconstructions of
paleo-sea levels, whether it is induced by tectonic events or not, is a science that takes advantage of the shape of modern
or fossil micro-atolls (Meltzner et al. 2006). However, we stress out that this study is not about reconstructing sea levels
using dead corals. Rather, we explained coral mortality using sea level data, primarily from altimetry data. The
agreement between altimetry and tide gauge data (Fig. 3) confirms that altimetry data are suitable to monitor sea level

variation close to a coast. More specifically, this confirms the value of using altimetry observations to help identifying the cause of shallow coral mortality, even without any other local *in situ* source of sea level data, as in Bunaken.

Interestingly, we found that sea level fall appeared to be responsible of coral mortality, while most recent climate change literature is generally focused on the present and future effects of sea level rise (Hopley, 2011). Geological records and present-time observation have already demonstrated that sea level variation is a driver of coral community changes. Sea level rise can have antagonistic effects: on the one hand, it can provide new growing space for corals. On the other hand, higher depth may enhance wave propagation and increase physical breakage in areas that were previously sheltered. If sea level rise is fast, corals may not keep up and the reef may be drowning relative to the new sea level. As such, sea level rise is seen as one of the three main climate change threats for coral reefs. This study reminds that the processes can be much more variable at ecological time-scale.

We aimed here to document the spatial scale and the cause of an ecological event that could be easily overlooked when documenting the 2016 El Niño impact on Indonesian coral reefs. Many studies have emphasized the role of hydrodynamics and sea level on the status and mortality of coral communities growing on reef flats (e.g., Anthony and Kerswell, 2007; Hopley, 2011; Lowe et al., 2016). Here we emphasize, with altimetry data for one the first time for a reef flat study (see Tartinville and Rancher 1997), that the 2015-2016 El Niño has generated such mortality, well before any ocean warming-induced bleaching. The exact time of the mortality remains unknown, but it is likely congruent to the lowest level in September 2015. The aspect of the colonies in February 2016, with algal turf covering the dead part (Fig. 1), is also consistent with a lowest sea level occurring few months earlier. The Figure 1 shows corals that were fine in May 2015 even when exposed to aerial exposure during low spring tide, without wave or wind, for several hours, during several days of spring tide. Thus, we assume the mortality was due to several weeks of lower water, including spring tide periods, which is compatible with the temporal resolution of the altimetry observations. The aerial exposure could have led to tissue heating, desiccation, photosystem or other cell functions damage (Brown, 1997). It is possible that colonies could have look bleached during that period (Brown et al. 1994). Lack of wind-induced wave in the September period also prevented wave washing and water mixing which could have limited the damage (Anthony and Kerswell, 2007).

The various satellite Sea Surface Temperature (SST) products for coral bleaching warning available at http://coralreefwatch.noaa.gov/ do not suggest any bleaching risk in the Bunaken region before June 2016, hence the wide mortality we observed can not be simply explained by ocean warming due to El Niño. We also verified on http://earthquake.usgs.gov/ that between the May 2015 habitat mapping survey and the February 2016 coral survey, no tectonic movement could generate such a 15 cm–uplift, with an upward shift of coral colonies relative to sea level as it has been reported in different places in the past, including in Sumatra, Indonesia after the 2004 Sumatra Earthquake (Meltzner et al. 2006). An uplift of that magnitude would be related to a significant event, but there are no reports higher than a 6.3 magnitude earthquake (16[th] September 2015, origin 1.884°N   126.429°E) in the north Sulawesi area for that period.

Altimetry data have been seldom used to study coral reef processes, even in a sea level rise era that may affect coral reef
communities and islands. They have been useful to assess the physical environment (wave, tide, circulation, lagoon
water renewal) around islands and reefs (e.g., Tartinville and Rancher, 1997; Andréfouët et al., 2001; Burradge et al.,
2003; Andréfouët et al., 2012; Gallop et al., 2014), or explain larval connectivity and offshore physical transport between
reefs (e.g., Christie et al., 2010), but this is the first time to our knowledge that altimetry data, including the new
SARAL/AltiKa data, are related to a coral ecology event. Different measures of sea level and sea level anomalies
confirmed an anomalous situation following the development of the 2015-2016 El Niño, resulting in lower sea level
regionally averaging 8 cm in the north of Sulawesi compared to the previous 4 years (Figs. 3-6). Mortality patterns on
coral colonies strongly suggests that sea level fall is responsible of the coral die-off that could reach 80% of reef flat
colonies that were in a keep-up position relative to, usually, rising sea-level in this region (Fenoglio-Marc et al; 2012).
While mortality due to sea level fall was characterized opportunistically in Bunaken NP, the impact remains unquantified
elsewhere. However, we speculate that similar events have occurred throughout the Indonesian seas when considering
ADT values for this region (Fig. 6). Particularly impacted by sea level fall could have been the stretch of reefs and
islands between South Sumatra, South Java, the Flores Sea and Timor, and the domain centered by Seram island and
comprised between East Sulawesi, West Papua and the Banda Sea. These areas have substantial reef flat presence (e.g.,
for the Lesser Sunda region comprised between Bali, Maluku and Timor islands, see maps in Torres-Pulliza et al., 2013).
Specifically for Bunaken NP, the event we have witnessed helps explaining long term observations of reef flat dynamics
and resilience. Indeed, our surveys and historical very high resolution satellite imagery show around Bunaken Island the
fast colonization of reef flats by *Heliopora coerula* and by carpets of branching *Montipora* in the years 2004-2012, a
period congruent to substantial rising sea level (Fig. 3) (Fenoglio-Marc et al. 2012). Rising seas has allowed these corals,
especially fast growing and opportunistic like *H. coerula* (Babcock, 1990; Yasuda et al., 2012) to cover previously bare
reef flats by taking advantage of the additional accommodation space. A similar process occurred in Heron Island reef
flats in Australia, with an artificially-induced sea level rise due to local engineering work (Scopélitis et al., 2011). In
Bunaken, and probably elsewhere in Indonesia and the Coral Triangle, the 2015-2016 El Niño event counter-balances
this period of coral growth with rising seas.
The ADT time-series (Fig. 4) suggests that similar low level situations have probably previously occurred, and almost
certainly at least in 1997-1998, the highest anomaly on altimetry record. Reef flat coral mortality reported in the Coral
Triangle as the consequences of bleaching in these years is thus most likely also the consequences of sea level fall. The
discrimination between thermal and sea level fall-induced mortality could be difficult to pinpoint on reef flats, if surveys
had occurred several months after the thermally-induced bleaching. In Bunaken NP, mortality due to sea level fall
preceded by nearly 7 months the first occurrences of bleaching in Indonesia, reported in April 2016. The real impact of
sea level fall could have been largely underestimated during all El Niño episodes and especially in Asia. The
implications for coral reef monitoring in the Coral Triangle are substantial. Surveys that may have started in April 2016
may be confused and assigned reef flat mortalities to coral bleaching. In future years, monitoring SLA may be as
important as monitoring SST. While there are several SST-indices specifically used as early-warning signals for potential
coral bleaching (Teneva et al., 2012), there are no sea level indices specific for coral reef flats. However, several ENSO
indices can help tracking the likelihood of similar events for Indonesia. The high correlation between the NINO3_4
index and ADT over the 1993-2016 period (monthly mean minus seasonal baseline, Fig. 6) shows this potential. Other
indices, such as the Southern Oscillation Index (SOI, computed as the pressure difference between Darwin and Tahiti),
or the Equatorial SOI (defined by the pressure difference between the Indonesia-SLP, standardized anomalies of sea
level, and the Equatorial Eastern Pacific SLP) appears to be even more suitable over Indonesia and the Coral Triangle to
develop suitable early-warning signals related to sea level variations.

## 5. Conclusion

This study reports coral mortality in Indonesia after a El Niño-induced sea level fall. The fact that sea level fall, or
extremely low tides, induces coral mortality is not new, but this study demonstrates that through rapid sea level fall, the
2015-2016 El Niño has impacted Indonesian shallow coral reefs well before that high sea surface temperature could
trigger any coral bleaching. Sea level fall appear as a major mortality factor for Bunaken Island in North Sulawesi, and
altimetry suggests similar impact throughout Indonesia. Our findings confirm that El Niño impacts are multiple and the
different processes need to be understood for an accurate diagnostic of the vulnerability of Indonesian coral reefs to
climate disturbances. This study also illustrates how to monitor local sea level to interpret changes in a particular coastal
location. For Indonesia coral reefs, in addition to sea level fall depending on the ENSO situation, further changes can be
expected, due to coral bleaching, diseases, predator outbreaks, storms and sea level rise (Baird et al., 2013; Johan et al.,
2014). Considering the level of services that shallow coral reefs offer, in coastal protection, food security and tourism,
the tools presented here offer valuable information to infer the proper diagnostic after climate-induced disturbances.

**Competing interests**

The authors declare that they have no conflict of interest.

**Acknowledgements**

This study was possible with the support of the INfrastructure DEvelopment of Space Oceanography (INDESO) project
in Indonesia, and its Coral Reef Monitoring Application. Fieldwork on Bunaken Island was authorized by the research
permit 4B/TKPIPA/E5/Dit.KI/IV/2016 delivered by the Ministry of Research, Technology and Higher Education of the
Republic of Indonesia to SA. This is ENTROPIE contribution 178.

308  .

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

Table 1: Mortality rates (mean ± standard deviation, n=6) of all corals for the 6 reef flat sites. The three dominant species were *Porites lutea*, *Heliopora*
*coerulea*, and *Goniastrea minuta*. Several species and genus were found only once. Standard deviation is not shown when only one measurement per type of
coral could be achieved (i.e., one colony per site).

| | | Coral | | | | | | | | | | |
|---|---|---|---|---|---|---|---|---|---|---|---|---|
| | | *Porites lutea* | *Heliopora* | *Goniastrea* | *Acropora* | *Galaxea* | *Cyphastrea* | *Montipora* | *Porites cylindrica* | *Lobophyllia* | *Pocillopora* | *Mean* |
| | 1 | 44±36 | 52±24 | | | | | | 42±40 | | | 46 |
| | 2 | 39±16 | 18±8 | 100±0 | | | | | | | 100 | 57 |
| **Site** | 3 | 54±5 | | 100±0 | 100±0 | 100 | 100 | | | | | 58 |
| | 4 | 20±17 | | 100 | 25 | | | | | 100 | | 55 |
| | 5 | 61±13 | 29±18 | | | 67 | | 100±0 | | | | 85 |
| | 6 | 52±23 | 70±8 | 46±51 | | 100 | | | | | | 47 |
| | *Mean* | *44* | *42* | *82* | *45* | *89* | *100* | *100* | *42* | *100* | *100* | *58* |



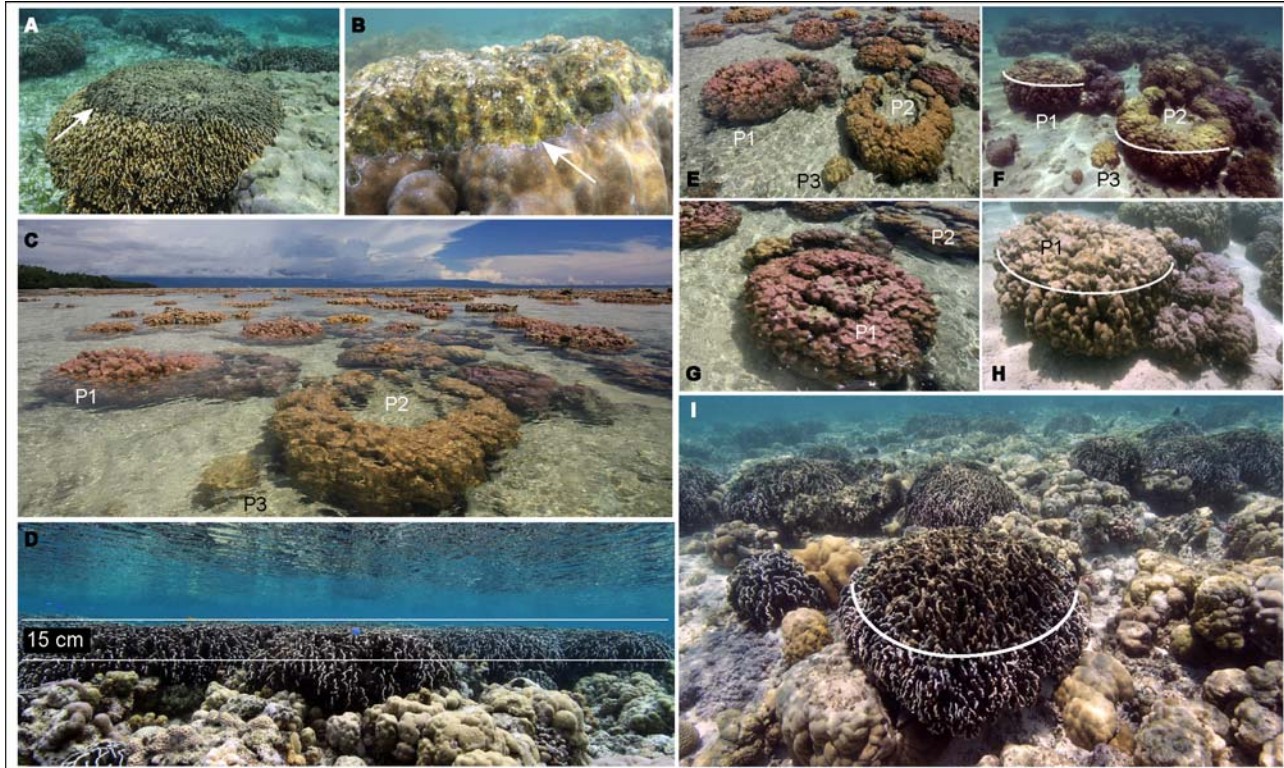

Figure 1: Bunaken reef flats. A: close-up of one *Heliopora coerula* colony with clear tissue mortality on the upper part of the colonies. B: same for a *Porites lutea* colony. C: reef flat *Porites* colonies observed at low spring tide in May 2014. Even partially above water few hours per month in similar conditions, the entire colonies were alive. D: a living *Heliopora coerula* (blue coral) community in 2015 in a keep-up position relative to mean low sea level, with almost all the space occupied by corals. In that case, a 15-cm sea level fall will impact most of the reef flat. E-H: before-after comparaison of coral status for colonies visible in C. In E, healthy *Porites lutea* (yellow and pink massive corals) reef flat colonies in May 2014 observed at low spring tide. The upper part of colonies is above water, yet healthy. F: Same colonies in February 2016. The white line visualizes tissue mortality limit. Large *Porites* colonies (P1, P2) at low tide levels in 2014 are affected, while lower colonies (P3) are not. G: P1 colony in 2014. H: viewed from another angle, the P1 colony in February 2016. I: Reef flat community with scattered *Heliopora* colonies in February 2016, with tissue mortality and algal turf overgrowth.

409

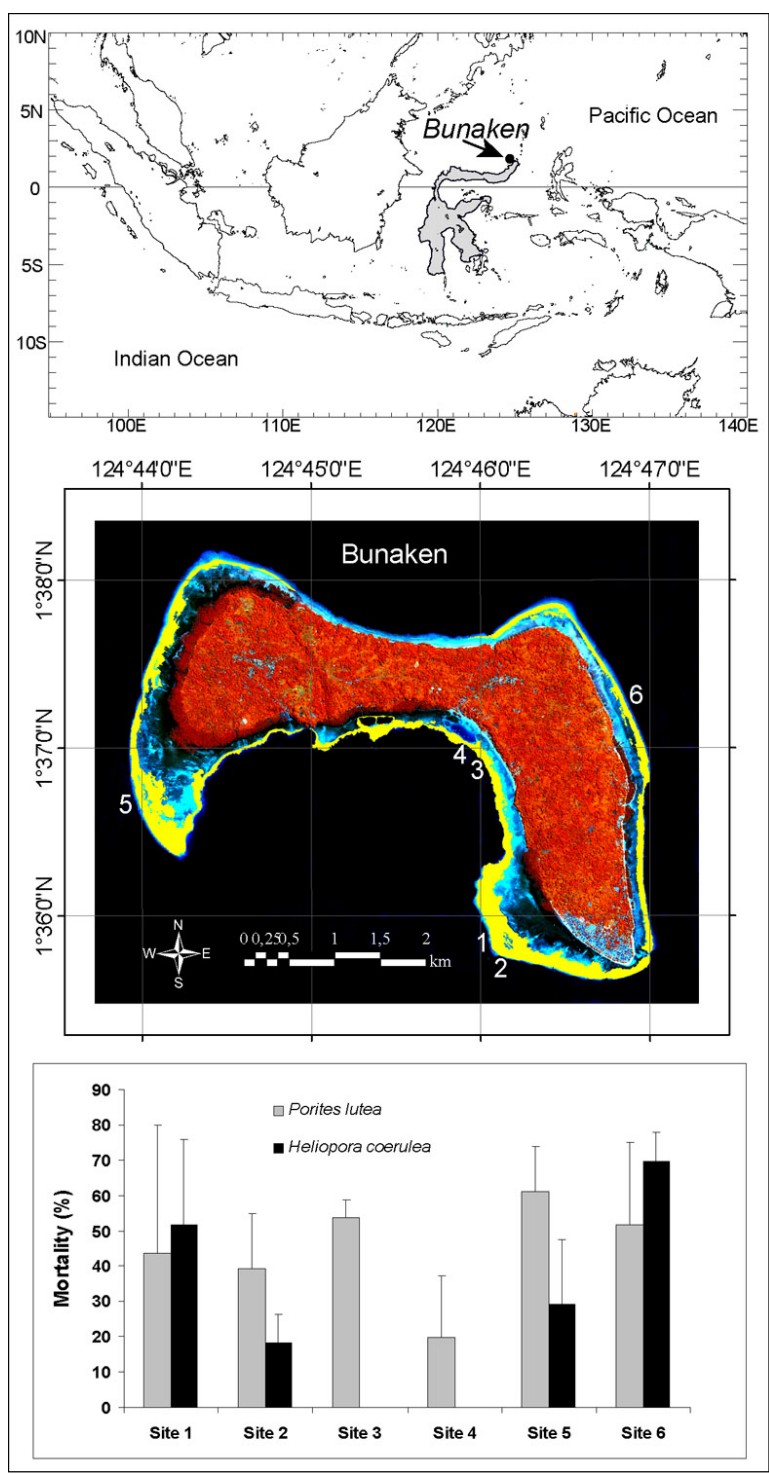

Figure 2: Top: Bunaken location in the north of Sulawesi, the large island in grey. Middle: Close-up of Bunaken Island. The yellow area shows where coral mortality occurred around Bunaken reef flats, with the position of six sampling stations. Dark areas between the yellow mask and the land are seagrass beds. Blue-cyan areas are slopes and reef flats without mortality. Bottom: Mortality rates for the 6 sites for two dominant species *Porites lutea* and *Heliopora coerulea*. The latter is not found on Sites 3 and 4. The number of colonies ranged between 10 and 30 per transect, depending on the size of the colonies.

417

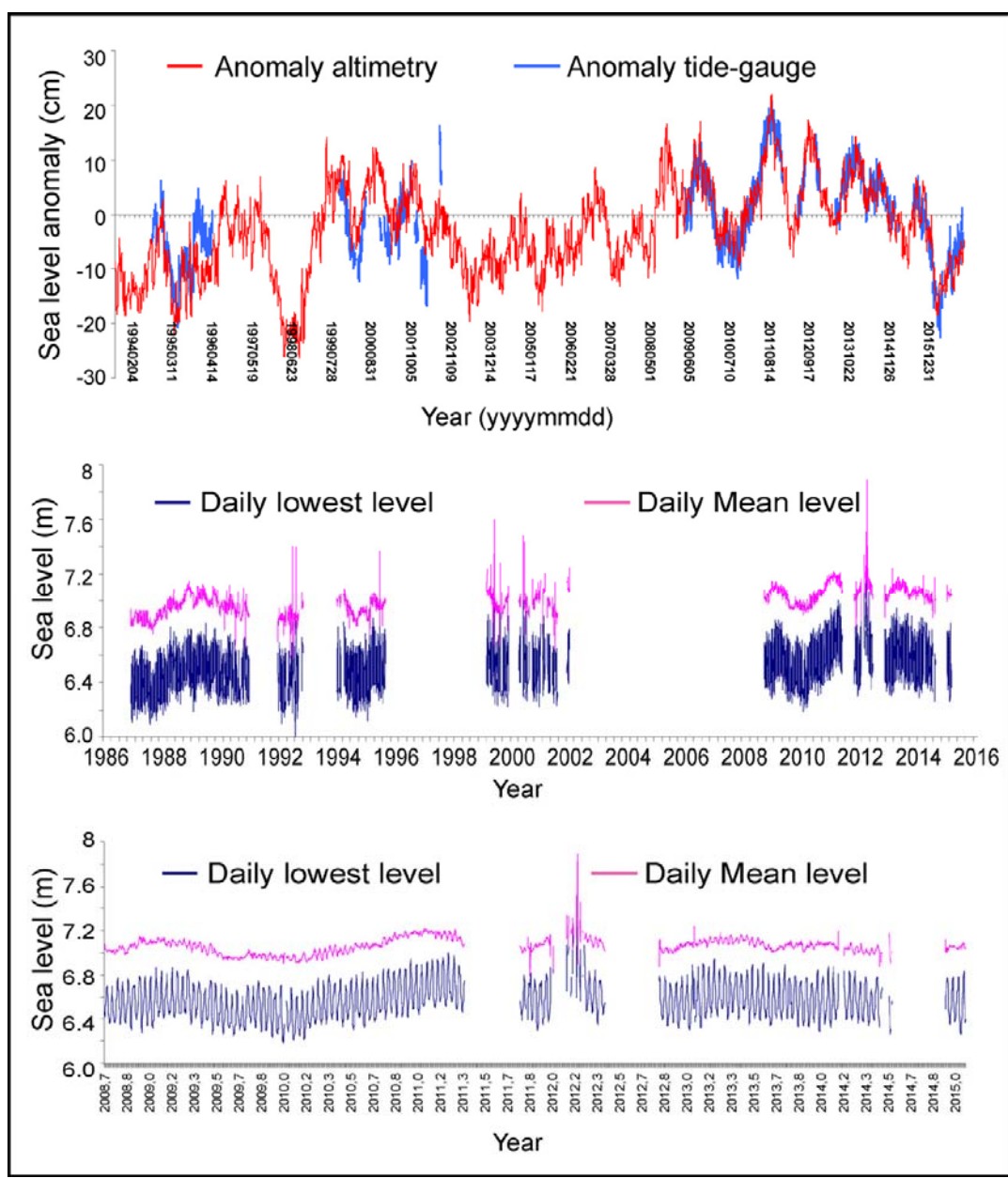

418

Figure 3: Sea-level data from the Bitung (east North Sulawesi) tide gauge, referenced against Bako GPS station. On top, sea level anomalies measured by the Bitung tide gauge station (low-quality data), and overlaid on altimetry ADT anomaly data for the 1993-2016 period. Note the gaps in the tide gauge time-series. Middle: Bitung tide gauge seal level variations (high-quality data, shown here from 1986 till early 2015) with daily mean and daily lowest values. Bottom, a close-up for the 2008-2015 period.

425

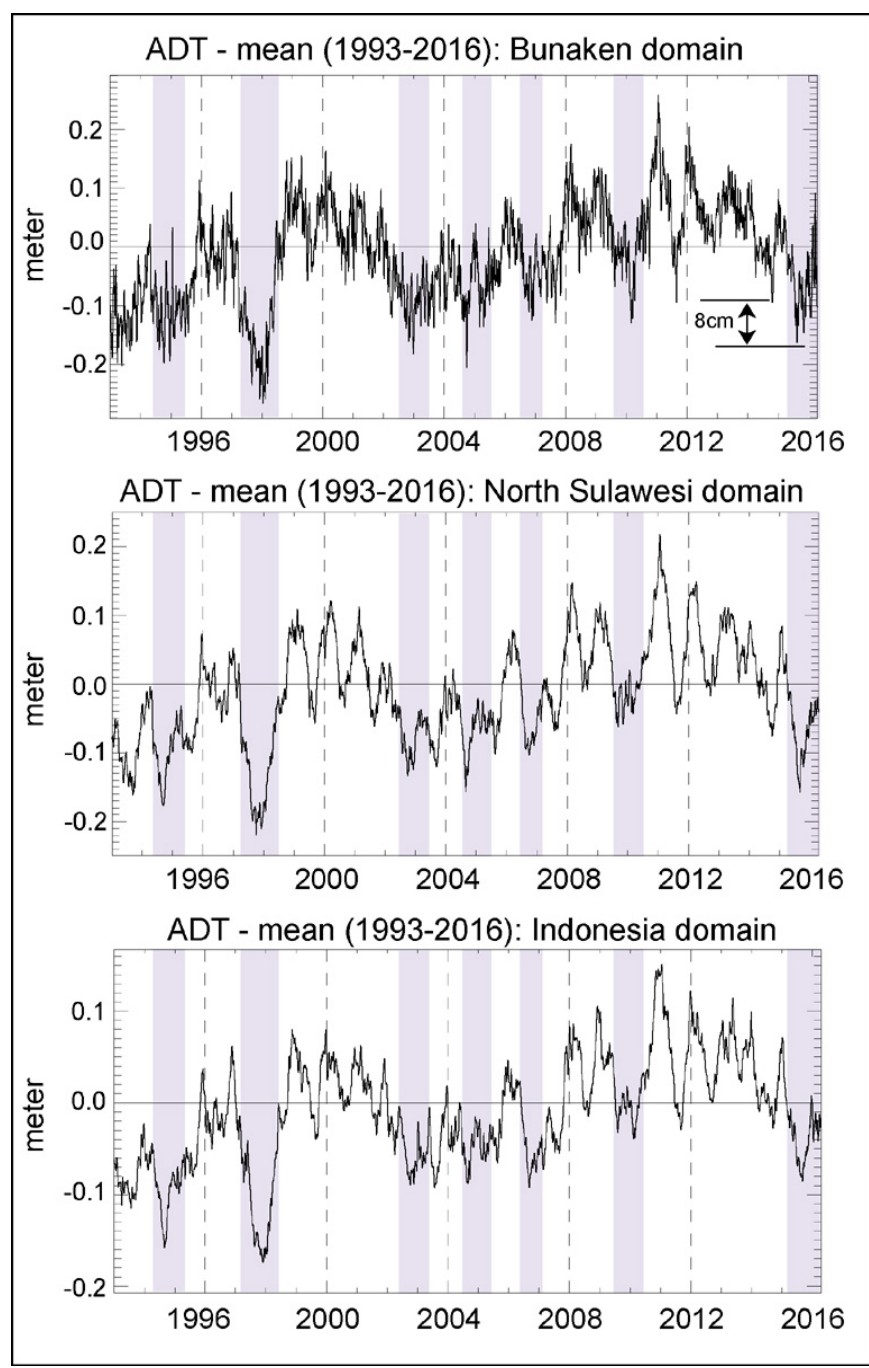

426

Figure 4: Time series of ADT, minus the mean over the 1993-2016 period, for Bunaken Island (top), North Sulawesi (middle), and Indonesia (bottom). The corresponding spatial domains are shown Figure 6. El Nino periods (http://www.cpc.ncep.noaa.gov/products/analysis_monitoring/ensostuff/ensoyears.shtml) are depicted with light shadings. The September 2015 minimum corresponds to a 8 cm fall compared to the minima the four previous years, and a 14 cm fall compared to the 1993-2016 mean. The 1998 El Niño displays the largest sea level fall.

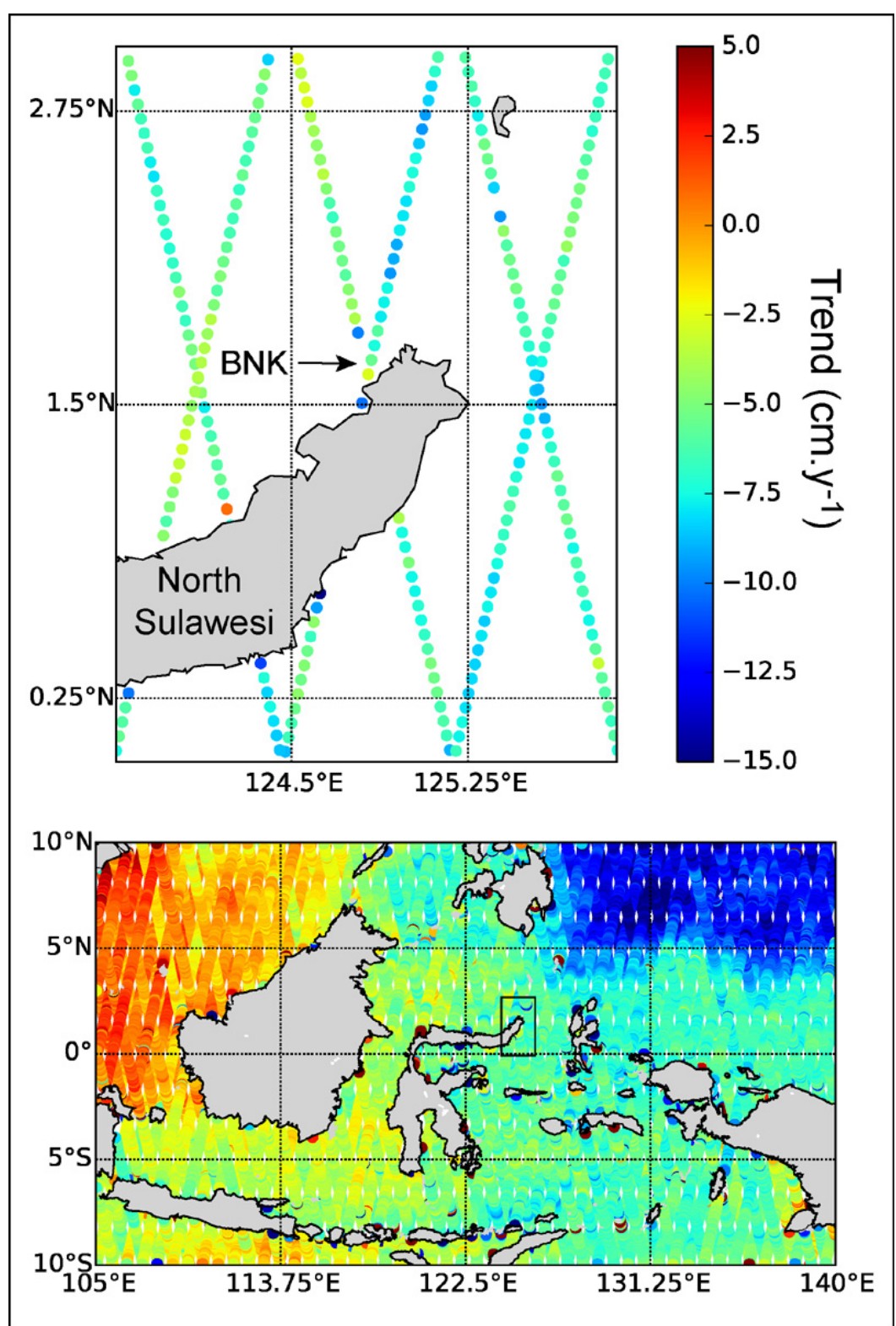

433

Figure 5: Top: Map of along-track SLA trend (in cm.year[-1]), 2013-2016, for the north Sulawesi area. The position of Bunaken Island is shown (BNK). Bottom: Map of along-track SLA trend (1-Hz), 2013-2016, for Indonesia. The domain on the top panel is the rectangle in the Indonesia map.

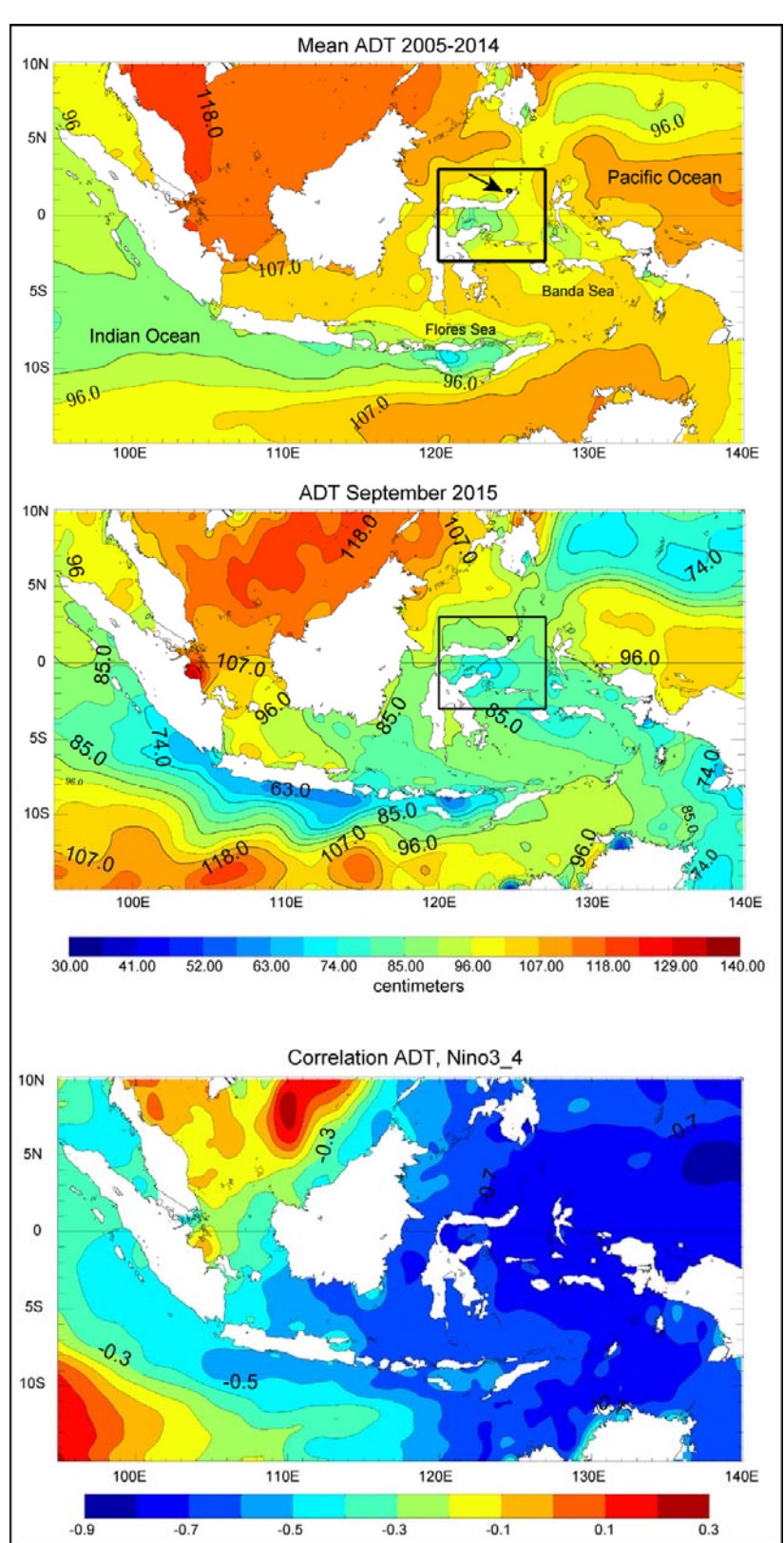

437

Figure 6: Top: Map of the 2005-2014 Absolute Dynamic Topography (ADT, in centimeters) average
over Indonesia. Middle: Map of the September 2015 ADT mean value over Indonesia. The two
squares indicate the domain just around Bunaken Island (arrow on top panel) and the north Sulawesi
domain used for the ADT time-series presented in Figure 4. Bottom: Map of correlation between
ADT and the Nino3-4 index (1993-2016, monthly average minus seasonal cycle).