# Peer review of "Coral mortality induced by the 2015-2016 El-Niño in Indonesia: 1"

_Biogeosciences, 2016_

## Referee Comment (RC1) · Anonymous Referee #1 · 28 Sep 2016

This paper presents the first report of coral mortality induced by El Niño related sea level fall event in 1995, and thus would contributes not only to coral reef science but also to ecosystem response to climate change studies in general.

The relation between sea level and the upper level of coral survival is a key to this study. However, the upper level of coral survival before this event is not quantitatively discussed, but only described as Mean Low Water (MLW). MLW which limit the coral survival before the event should be quantitatively shown in the time series ADT in Figure 3, and then the mortality of the upper 15cm of corals should be defined in sea level changes.

Figure 2 shows coral mortality area spatially. To obtain spatial distribution of coral

mortality, the authors states that "reef flats were visually surveyed and recent mortality was recorded. Geographic coordinates of the presence of mortality were compiled to map its extent". However, it seems to me the area was not only mapped by field survey, but also by remote sensing. If so, the authors should describe how remote sensing was applied to map the mortality area.

In 1997-1998, sea level dropped lower than the event in 2016. The corals should also have recorded the event in 1997-1998. The coral of P2 in Figure 1 forms a microatoll with a depression, which seems to be a record of sea level drop in 1997-1998. Twenty years passed since then, and horizontal growth during this period might be 20 cm. If so, the authors' observation shows coral microatolls record not only tectonic relative sea level changes, but also climate related sea level changes. Further discussion of this point referring to reconstruction of tectonic sea level change by coral microatolls would extend viewpoint of this paper.

The authors reconstruct sea level changes by satellite remote sensing. Reliable sea level history is obtained by tidal gauge. The remote sensing reconstruction should be compared with tide gauge data in this area.

---

## Referee Comment (RC2) · Anonymous Referee #2 · 25 Nov 2016

General comments

This is a very interesting observation that describes substantial mortality among reef-flat corals in Indonesia that occurred during the intense El Nino of 2015-16. The authors provide a convincing case that the mortality, confined to a distinct band across the upper portions of the coral colonies, was caused by extended sub-aerial exposure during anomalous low mean sea level some months before the peak sea temperatures. The distinctness of the band and the absence of death among marginally deeper corals is very convincing evidence that exposure to air, not hot water, was the cause of coral death. Of itself, this is not such a profound observation but what makes the paper's cause-effect hypothesis plausible is the inclusion of the sea level data, which shows both the vertical extent of the 2015 low sea-level event, its widespread occurrence across Indonesia, and precedents in 2005 and 1998. It surprised me the authors did not make a bit more of the paradox that one of the big 3 global climate change on coral reefs is meant to be a rise in sea-level. It is also likely that the dead patch on P2 is a scar of the previous sub-aerial exposure of this colony in 2005, and I was surprised this was not mentioned.

Specific comments

While I found the overall story plausible and interesting, there is a key flaw and that is the quality of the 6 part colour plate that constitutes the major biological evidence for the story. I don't think that even high quality final prints will show the band of coral death very clearly and I suggest the authors need to include some close-up panels that clearly show the difference between the living and dead surfaces of P1, P2 and P3 and in the Heliopora coerulea colony. Table 1 cites the percentage and variability of mortality among corals, but I think it will need, in each cell, the numbers or colonies on which the percentages are based, to satisfy readers with a more quantitative bent. Figure 2 could do with a line drawing of that bit of the N Sulawesi coast to better present the geographic setting for the false colour map of the Bunaken reef. Make the present map of all Indonesia an inset to the N Sulawesi map.

The reference to other literature is quite limited. The regional sea-level story seems like a major oceanographic finding. Has it been described in detail elsewhere, and is it controversial? Also the biological credibility of the author's interpretation would be strengthened by more reference to the literature. For example, Wellington and Glynn (2007) have a plate of a similar pattern of death on large corals at 6 m depth, and BE Brown and co-workers describe similar patterns on the reef-flat at Phuket.

Wellington GM, Glynn PW (2007) Responses of coral reefs to ENSO sea warming events. In Aronson, RB. Geological approaches to coral reef ecology, Springer, 345 – 385

Brown et al, (2002) Experience shapes the susceptibility of a reef coral to bleaching. Coral Reefs 21; 119 – 126.

Please also note the supplement to this comment:
http://www.biogeosciences-discuss.net/bg-2016-375/bg-2016-375-RC2-supplement.pdf

———————————————————————

[Figure]

**Supplement:**

[revised manuscript text omitted]

---

## Author Comment (AC1) · 21 Dec 2016

**Dear Editor**

We thank the two reviewers for their comments. Both reviewers appear to be positive. Their comments aimed to clarify some aspects of the context of the work, and some aspects of the methods. We have tried to comply with all the requests. We provide hereafter explanations and additional information, that can be used for the revised version, or that can remain available in the Discussion archive.

**Specifically,**

Reviewer 1 requested:

1. a clarification of the relationship between Mean Low Water (MLW), which indicates the upper level of coral survival, and the ADT anomaly (Figure 3 of the initial manuscript). The reviewer said "However, the upper level of coral survival before this event is not quantitatively discussed, but only described as Mean Low Water (MLW). MLW which limit the coral survival before the event should be quantitatively shown in the time series ADT in Figure 3, and then the mortality of the upper 15cm of corals should be defined in sea level changes".

Response: The request can not be fulfilled since we do not have precise measurements of Mean Low Water level for Bunaken. For this, data from a pressure sensor are ideally needed. Instead, tide gauge data could be used, but tide-gauge data are scarce in Indonesia. Fortunately there are two tide-gauges in the north of Sulawesi in the city of Bitung, east of Bunaken, by latitude 1.430N and longitude 125.200E on the other side of Sulawesi compared to Bunaken, our study site. Thus, while tide-gauge data are available in the region, they are not exactly on Bunaken, but can help visualize the range of conditions found in Bunaken.

Bitung data can be retrieved from the Sea Level Center in Hawaii (SLCH), specifically at <a href="http://uhslc.soest.hawaii.edu/thredds/uhslc\_quality\_daily.html?dataset=RQD033A">http://uhslc.soest.hawaii.edu/thredds/uhslc\_quality\_daily.html?dataset=RQD033A</a>

The Sea Surface Height (SSH) provided is referenced, for Bitung, against a GPS station located at Bako (http://www.igs.org/igsnetwork/network by\_site.php?site=bako) which is itself referenced against the WGS84 ellipsoid. Hence, raw Bitung SSH do not represent absolute depth above the Bitung seafloor. SLCH provides high quality data (available till early 2015) that have been controlled for most outliers and errors, and lower quality data that includes the most recent coverage, included our period of interest (2015-2016).

We include below a graph showing from the available Bitung data the daily mean sea level (that can be compared to sea level as provided by altimetry), and the daily lowest level (not directly measured by altimetry) (Fig. 1). This graph only aims to show what was likely happening in Bunaken before El Niño in terms of range of sea level variations due to normal tide fluctuations. The daily lower value exhibits a ~40-cm variation from neap tide to spring tide. The limit of coral survival is species dependent, but close to the mean of the daily lowest level (blue curve). If this mean is changing through time, the limit of mortality also changes dynamically. Using the mean low level for the entire period of data coverage could be misleading to characterize the situation for a precise period. In 2014, and 2015, we witnessed during spring tide conditions *Porites* corals that had the upper part of the colonies well above the sea level, and without signs of mortality (see Picture 1). Hence, the upper limit of coral survival is somewhere around 20 cm above the spring tide lowest level for the end of the period shown on the graph below.

Figure 1 : Available data from the Bitung (east North Sulawesi) tide-gauge referenced against Bako GPS station. On top, the entire available data, from 1986 till 2015. Bottom, a close-up of the 2008-2015 period.

---

## Author Comment (AC2) · 21 Dec 2016

Dear Editor

We thank the two reviewers for their comments. Both reviewers appear to be positive. Their comments aimed to clarify some aspects of the context of the work, and some aspects of the methods. We have tried to comply with all the requests. We provide hereafter explanations and additional information, that can be used for the revised version, or that can remain available in the Discussion archive.

Specifically,

Reviewer 1 requested:
1. a clarification of the relationship between Mean Low Water (MLW), which indicates the upper level of coral survival, and the ADT anomaly (Figure 3 of the initial manuscript). The reviewer said "*However, the upper level of coral survival before this event is not quantitatively discussed, but only described as Mean Low Water (MLW). MLW which limit the coral survival before the event should be quantitatively shown in the time series ADT in Figure 3, and then the mortality of the upper 15cm of corals should be defined in sea level changes*".

Response: The request can not be fulfilled since we do not have precise measurements of Mean Low Water level for Bunaken. For this, data from a pressure sensor are ideally needed. Instead, tide gauge data could be used, but tide-gauge data are scarce in Indonesia. Fortunately there are two tide-gauges in the north of Sulawesi in the city of Bitung, east of Bunaken, by latitude 1.430N and longitude 125.200E on the other side of Sulawesi compared to Bunaken, our study site. Thus, while tide-gauge data are available in the region, they are not exactly on Bunaken, but can help visualize the range of conditions found in Bunaken.

Bitung data can be retrieved from the Sea Level Center in Hawaii (SLCH), specifically at http://uhslc.soest.hawaii.edu/thredds/uhslc_quality_daily.html?dataset=RQD033A
The Sea Surface Height (SSH) provided is referenced, for Bitung, against a GPS station located at Bako (http://www.igs.org/igsnetwork/network_by_site.php?site=bako) which is itself referenced against the WGS84 ellipsoid. Hence, raw Bitung SSH do not represent absolute depth above the Bitung seafloor. SLCH provides high quality data (available till early 2015) that have been controlled for most outliers and errors, and lower quality data that includes the most recent coverage, included our period of interest (2015-2016).

We include below a graph showing from the available Bitung data the daily mean sea level (that can be compared to sea level as provided by altimetry), and the daily lowest level (not directly measured by altimetry) (Fig. 1). This graph only aims to show what was likely happening in Bunaken before El Niño in terms of range of sea level variations due to normal tide fluctuations. The daily lower value exhibits a ~40-cm variation from neap tide to spring tide. The limit of coral survival is species dependent, but close to the mean of the daily lowest level (blue curve). If this mean is changing through time, the limit of mortality also changes dynamically. Using the mean low level for the entire period of data coverage could be misleading to characterize the situation for a precise period. In 2014, and 2015, we witnessed during spring tide conditions *Porites* corals that had the upper part of the colonies well above the sea level, and without signs of mortality (see Picture 1). Hence, the upper limit of coral survival is somewhere around 20 cm above the spring tide lowest level for the end of the period shown on the graph below.

[Figure]

Figure 1 : Available data from the Bitung (east North Sulawesi) tide-gauge referenced against Bako GPS station. On top, the entire available data, from 1986 till 2015. Bottom, a close-up of the 2008-2015 period.

[Figure]

Picture 1: Reef flat close to lowest spring tide the 15[th] May 2014. The upper part of *Porites* colonies are fine and can obviously accommodate these conditions for few hours per month. The limit of growth is higher (~20 cm) than the lowest spring tide limit. This picture is from the same location as the Figure 1 on the manuscript, taken from a lower angle.

Therefore, the ~15cm fall that we observed on altimetry data around Bunaken and on most of east Indonesia changed for a short time (of several weeks) the lowest levels, and these changes lasted long enough so that coral tissues were damaged by too much UV and air

exposure. This fall shifted during few weeks in August-September 2015 the mean low level close to the pre-El Niño lowest levels shown in the Figure 1 above.

The request *"the mortality of the upper 15cm of corals should be defined in sea level changes"* is unclear. Not all corals have a 15cm band of tissue mortality, as it depends on the depth of the coral. See the new photographs that we have added in response to Reviewer 2 (Picture 2). We hope that the above graph and discussion replies to this comment.

2. an explanation on how the extent of the mortality was mapped. Reviewer 1 said: *"Figure 2 shows coral mortality area spatially. To obtain spatial distribution of coral mortality, the authors states that "reef flats were visually surveyed and recent mortality was recorded. Geographic coordinates of the presence of mortality were compiled to map its extent". However, it seems to me the area was not only mapped by field survey, but also by remote sensing. If so, the authors should describe how remote sensing was applied to map the mortality area."*

Response: Indeed, the map of mortality (Figure 2, middle, of the submitted paper) is created using a pre-existing habitat map from remote sensing (described in Ampou 2016, a PhD thesis recently defended $6^{th}$ December 2016). Our survey was done for all coral habitats along the reef. If we recorded mortality for that habitat, we flagged the corresponding polygon in the map as a mortality area. Hence, the yellow mask in Figure 2 is the mask corresponding to all mapped habitats where mortality was found, but mortality itself is not inferred from remote sensing.
We have clarified this in the new Material and Methods section.

3. further discussion related to reconstruction of sea level using micro-atolls. The reviewer said *"Further discussion of this point referring to reconstruction of tectonic sea level change by coral microatolls would extend viewpoint of this paper. »*

Response: Indeed, there is a common ground between our observations and the use of coral to reconstruct sea level. Reconstructions of paleo-sea level, whether it is induced by tectonic relative sea level change or not, is a science that takes advantage of the shape of modern or fossil micro-atolls. We now briefly point out this aspect in the Introduction. However, we stress out that our study is clearly not about reconstructing sea levels using dead corals. Rather, we explain coral mortality using sea level data.

4. comparison of tide gauge data and altimetry sea level data. The reviewer said: *"The authors reconstruct sea level changes by satellite remote sensing. Reliable sea level history is obtained by tidal gauge. The remote sensing reconstruction should be compared with tide gauge data in this area."*

Response: We show below publicly available sea level data from the aforementioned Bitung tide-gauge (low quality data to include the end of 2015 and 2016 period) to show the agreement between altimetry data and tide gauges (Fig. 2). The Bitung SSH data and the Sea Level altimetry data are comparable because we show the anomalies relative to the mean for the same period of concurrent measurement.
Altimetry data are extracted from the AVISO server: http://misgw-sltac.vlandata.cls.fr:45080/thredds/dodsC/dataset-duacs-rep-global-merged-allsat-msla-l4

These are interpolated data, from various altimetry missions. The selected retrieved location is the closest available from Bitung (1.375N and longitude 125.125E). Data are available without interruption since 1993.

[Figure]

Figure 2: Sea level anomalies measured by the Bitung tide gauge station, and by altimetry (AVISO data server. Note the gaps in the tide gauge time-series. The anomaly was computed for both data sets by taking into account only the periods concurrently monitored by both types of measurements.

The good correlation (Pearson r=0.83) and the agreement in anomalies confirm that altimetry data are perfectly suitable to monitor sea level variation close to a coast. This confirms the value of using altimetry observations even without other local source of sea level data, as in Bunaken, to identify the cause of coral mortality (sea level fall).

Reviewer 2 suggests:
5. to emphasize the paradox between our observations and the fact that one of the three main climate change threats for coral reefs is sea level rise.

Response: this can be now done at the beginning of the new Discussion. The following sentence will be added: "*Geological records and present-time observation have demonstrated that sea level variation is a driver of coral community changes. Sea level rise can have antagonistic effects: on the one hand, it can provide new growing space for corals. On the other hand, higher depth may enhance wave propagation in areas that were previously sheltered and increase coral physical breakage. If sea level rise is fast, corals may not keep up and the reef may be drowning relative to the new sea level. As such, sea level rise is seen as one of the three main climate change threats for coral reefs*"

6. enhancing the color plate with close-up of mortality on colonies.

Response: we have modified the color plate Figure 1 to include the two following close-up pictures with clear banding due to tissue mortality, for a *Heliopora* colony (Picture 2, left) and for a *Porites* colony (right). More information has also been provided on the Figure 1 caption.

[Figure]

Picture 2: Bands of dead tissue as seen in February 2016. Left: *Heliopora coerula* colony; right: A *Porites lutea* colony.

7.  adding the number of colonies surveyed for each transect and station

Response: The information we have provided is the percentage of mortality found on all colonies on the transects, but we did not keep the information on how many colonies were measured. Photographs of the transects are available but they do not cover the entire transects. The number of colonies on 20m long transect is however high since coral cover was high. We estimate the number of colonies ranged between 10 and 30 per transect, depending on the size of the colonies. This estimation has been provided in the new caption of the Figure 2.

8.  reorganize the map. The reviewer said: "*Figure 2 could do with a line drawing of that bit of the N Sulawesi coast to better present the geographic setting for the false colour map of the Bunaken reef. Make the present map of all Indonesia an inset to the N Sulawesi map.* »

Response: The Figure 2 has been re-organized as suggested.

9.  suggest some additional references

Response: while we did not cite the 2 references provided by the Reviewer, we have previously included similar references to illustrate the same facts, including references by Brown et al. on Phuket reefs. However, we have now added the 2 suggested references.

In addition, Reviewer 2 provided edits on the draft itself, for English corrections that were all kept, if the initial text was not removed to accommodate the comments above.

END.